# Bridging the Reality Gap: A Benchmark for Physical Reasoning in General World Models with Various Physical Phenomena beyond Mechanics

## Abstract

While general world models have demonstrated excellent capability in modeling and simulating the world through video understanding and generation, their ability to reason about physical phenomena beyond mechanics remains underexplored. This includes crucial aspects like thermodynamics, electromagnetism, and optics, all of which are fundamental for simulating and predicting real-world dynamics. Existing benchmarks for evaluating physical reasoning in models often rely on datasets consisting solely of simulator-generated, virtual videos, limiting their generalizability to real-world scenarios. This limitation hinders the comprehensive evaluation of general world models' physical reasoning in real-world scenarios. To bridge this gap, we introduce the Physics-RW benchmark, a physical reasoning dataset constructed from real-world videos. Encompassing a broad spectrum of real-world phenomena—mechanics, thermodynamics, electromagnetism, and optics—Physics-RW offers a comprehensive evaluation platform. We conducted extensive experiments on the Physics-RW benchmark, and the results indicate that there is still significant room for improvement in the physical reasoning abilities of general world models. We further analyzed the experimental results and explored several avenues for improvement. Virtual environment finetuning and physical knowledge injection via prompts demonstrate the potential for enhancing zero-shot physical reasoning ability [1].

## 1 Introduction

Equipping artificial intelligence with the ability to comprehend and predict the dynamic nature of the world is crucial for advancing artificial general intelligence. With the emergence of Sora (Brooks et al., 2024), its outstanding generative ability highlights the potential of the general world models in simulating and modeling the world (Brooks et al., 2024; Zhu et al., 2024; Liu et al., 2024b). As physics is one of the challenges in modeling the world (Poupyrev et al., 2024), the general world models are expected to have the ability to understand the physical phenomena present in the real world (Wang et al., 2024; Cho et al., 2024).

The real world abounds in diverse physical phenomena, encompassing mechanics, thermodynamics, electromagnetism, optics, and more. For instance, the apparent bending of chopsticks immersed in water is not due to their deformation, but rather a light refraction phenomenon. Understanding such phenomena is crucial for interpreting and generating video content. While recent models like Sora (Brooks et al., 2024) and WorldDream (Wang et al., 2024) exhibit some capability in this area, the extent of their understanding and their ability for accurate physical reasoning remains unclear.

Currently, efforts have been made to construct benchmarks for evaluating physical reasoning in deep learning-based models. They design tasks from different perspectives, such as intuitive physics (Riochet et al., 2021), object motion (Bakhtin et al., 2019; Yi et al., 2020; Bear et al., 2021), physical properties of objects (Chen et al., 2021; Tung et al., 2023), and dynamic interactions (Li et al., 2023c), then use simulators or physics engines to generate video datasets or corresponding game

---

[1]The data and source code of our work will be publicly available on GitHub once the manuscript is accepted.

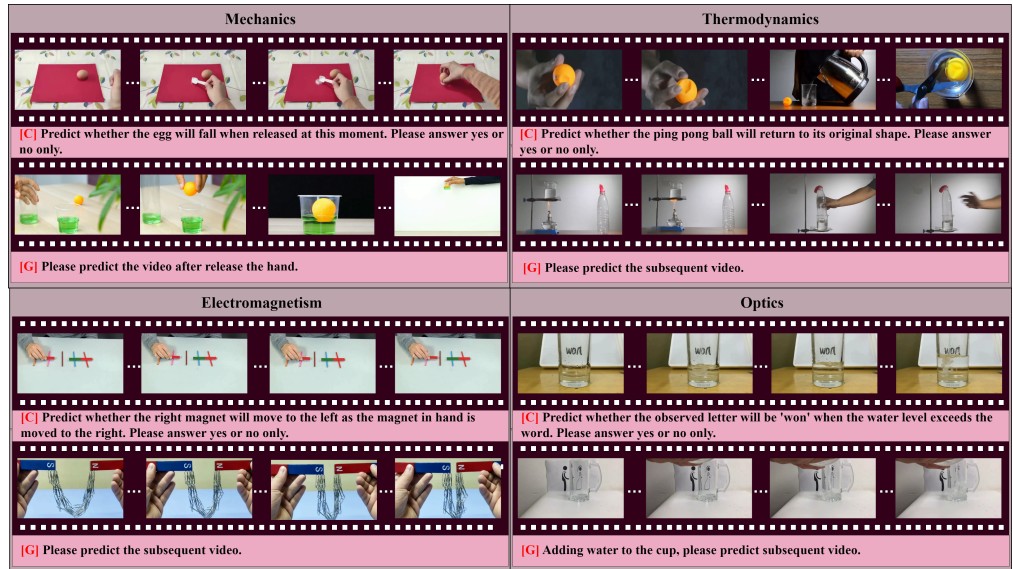

Figure 1: Physics-RW benchmark covers four categories of physical phenomena reasoning tasks, namely mechanics, thermodynamics, electromagnetism, and optics. Each task comprises two evaluation types: classification and video generation. [C]/[G] respectively indicate the instruction for classification and video generation task types.

Table 1: Comparison between Physics-RW and prior physical reasoning benchmarks. "Diverse Scenarios" refers to the benchmark covering multiple scenarios or physical characteristics. For example, our mechanics phenomena tasks include elasticity, gravity, resistance, etc., while our optics phenomena tasks include light reflection and refraction, etc. "Diverse Phenomena" denotes the benchmark covering multiple categories of classical physics. For example, our benchmark includes four kinds of physical phenomena, i.e., mechanics, thermodynamics, electromagnetism, and optics.

| Benchmarks | Diverse Scenarios | Real-World | Diverse Phenomena | Zero-shot Inference |
|---|---|---|---|---|
| PHYRE Bakhtin et al. (2019) | ✓ | ✗ | ✗ | ✗ |
| CLEVERER Yi et al. (2020) | ✓ | ✗ | ✗ | ✗ |
| Physion Bear et al. (2021) | ✓ | ✗ | ✗ | ✗ |
| Comphy Chen et al. (2021) | ✓ | ✗ | ✓ | ✗ |
| Physion++ Tung et al. (2023) | ✓ | ✗ | ✗ | ✗ |
| I-PHYRE Li et al. (2023c) | ✓ | ✗ | ✗ | ✓ |
| Physics-RW | ✓ | ✓ | ✓ | ✓ |

scenes. However, these prior benchmarks suffer from limitations: (1) Limited Generalizability to Reality: Datasets generated by simulators may not fully reflect the ability of models to understand real-world physical phenomena. (2) Restricted Scope: The focus mainly on object motion and collision limits the assessment of broader physical phenomena encountered in real-world scenarios. (3) Limited Zero-shot Evaluation: Most benchmarks evaluate performance based on training on specific datasets, neglecting zero-shot capabilities.

Naturally, an urgent research question arises: "**Can general world models perform accurate physical reasoning across various phenomena?**" To answer this, we constructed a comprehensive physical evaluation benchmark, named Physics-RW, based on real-world video-language data to validate the physical reasoning capability of general world models. Encompassing the four major categories of classical physics (mechanics, thermodynamics, electromagnetism, and optics), Physics-RW is designed for vision-language physical reasoning tasks. As illustrated in Figure 1, each physical phenomenon category includes two task types: classification and video generation. In the classification task, models perform physical inference by answering yes/no questions based on a given video clip. The video generation task type requires models to generate subsequent videos that demonstrate their understanding of the physical phenomena.

We conducted extensive experiments based on the Physics-RW benchmark to assess the zero-shot physical reasoning ability of general world models in real-world scenarios. The experimental results reveal that current general world models exhibit relatively limited physical reasoning capabilities,

highlighting significant room for improvement. We further analyze the experimental results to provide some insights for future works.

Our contributions are three-fold:

- We introduce a new benchmark, named Physics-RW, to address the current need for evaluating the physical reasoning ability of general world models in real-world scenarios. Physics-RW benchmark encompasses the majority of physical phenomena found in classical physics (i.e., mechanics, thermodynamics, electromagnetism, and optics) and stands as the pioneering benchmark constructed from real videos (§3).

- We conducted a comprehensive assessment of the physical reasoning capabilities of general world models on Physics-RW benchmark in a zero-shot manner, revealing that current models exhibit limited ability to infer real-world physical phenomena (§4).

- We further conducted an in-depth analysis, explored several avenues for improvement, and anticipate offering potential solutions (virtual environment finetuning and physical knowledge injection via prompts) for future works (§5).

## 2 RELATED WORK

### 2.1 VLM AS GENERAL WORLD MODELS

Large language models (LLMs) such as GPT-4 (Achiam et al., 2023), LLaMA-2 (Touvron et al., 2023), Qwen (Bai et al., 2023), etc., encode vast amounts of text data to learn world knowledge (Gurnee & Tegmark, 2023). Compared to text, videos can provide more detailed information, which is crucial for modeling physical phenomena in the world. Jointly modeling videos and text can help models better understand and represent the world, aligning with the goal of constructing a general world model (Liu et al., 2024a; Yang et al., 2024). Currently, visual language models (VLM), e.g., Video-ChatGPT (Maaz et al., 2023), Video-LLaMA (Zhang et al., 2023a), LWM (Liu et al., 2024a), Sora (Brooks et al., 2024), and Latte (Ma et al., 2024), have demonstrated outstanding performance in video understanding and video generation tasks. However, the physical reasoning ability of general world models, as one of their key capabilities to simulate the world, has yet to undergo comprehensive evaluation in real-world scenarios.

### 2.2 PHYSICAL REASONING BENCHMARKS

Given humans' ability to grasp physical concepts and perform physical reasoning, empowering artificial intelligence systems with this capability has garnered significant research attention (Qi et al., 2020; Duan et al., 2021; Li et al., 2022; Wu et al., 2022; Duan et al., 2022). To comprehensively evaluate these models' physical reasoning abilities, various physical reasoning benchmarks have been proposed (Bakhtin et al., 2019; Girdhar & Ramanan, 2019; Baradel et al., 2019; Riochet et al., 2021; Bear et al., 2021; Patel et al., 2022; Melnik et al., 2023; Li et al., 2023c; Tung et al., 2023). However, previous benchmarks typically focus on the movement and collisions of objects, with a relatively narrow range of physical phenomena. Additionally, their datasets are often generated by simulators or physics engines, making it difficult to comprehensively evaluate the physical reasoning ability in real-world scenarios. In contrast to them, our proposed Physics-RW benchmark is constructed from real-world videos, encompassing mechanics, thermodynamics, electromagnetism, and optics phenomena, covering most types of physical phenomena found in the real world. The comparison between Physics-RW benchmark and prior physical reasoning benchmarks is presented in Table 1.

## 3 BENCHMARK DESIGN

### 3.1 DATA CONSTRUCTION

We constructed a real-world physical reasoning benchmark by collecting videos based on four categories of physical phenomena: mechanics, thermodynamics, electromagnetism, and optics. To achieve this, we employed a two-pronged approach:

**Web-based Video Collection.** We established relevant keyword sets for each phenomenon category (e.g., mechanics, thermodynamics, electromagnetism, and optics) and used them to search for videos online. For example, mechanics keywords included elasticity, gravity, resistance, buoyancy, etc, while thermodynamics keywords encompassed thermal expansion and heat transfer, etc. Electromagnetism leveraged keywords like electromagnetic induction, demagnetization, and magnet experiments, and optics utilized terms like reflection of light, refraction of light, and propagation of light.

**Controlled Video Filming.** In addition to the web-based collection, we also shot our own domino collision videos, featuring variations in domino length, combination, and the number of dominoes impacting the target object. These controlled videos were categorized under the mechanics category.

Following video data collection, we meticulously segmented and selected video portions that explicitly showcased the targeted physical phenomena.

## 3.2 TASK CONSTRUCTION

As our benchmark incorporates two task types (i.e., classification and video generation) for each phenomenon, we transformed these video segments into specific task types:

**Classification Task.** We manually created instruction-answer pairs based on the segments. To facilitate quantitative evaluation, we incorporated prompts within the instructions, guiding the model to respond with a simple "yes" or "no" answer.

**Video Generation Task.** Segments were further split into two parts. We expect the model to generate the subsequent video (second part) based on the first part of the video and the provided instructions.

Finally, we constructed instructions in both Chinese and English, enabling future evaluations of world models designed for these two languages.

## 3.3 EVALUATION METRICS

To conduct quantitative performance statistics, we adopt different metrics for each task type. Specifically, for the classification task type, we use the commonly used metrics of Accuracy (ACC) and F1-score (F1) as evaluation metrics. For the video generation task type, we use FVD to evaluate the quality of the generated video. The smaller the FVD is, the higher the video quality will be.

## 3.4 DATASET STATISTICS

After manual annotation and verification, we obtained datasets corresponding to four categories of physical phenomena. The amount of data collected and annotated for each category varies due to differing phenomenon scopes and the challenges associated with data collection and annotation. Table 2 presents the data statistics, where T1, T2, T3, and T4 correspond to the mechanics, thermodynamics, electromagnetism, and optics tasks, respectively.

## 4 EXPERIMENTS

### 4.1 SELECTED MODELS

We select the latest and representative baseline models for different task types in the Physics-RW benchmark. Specifically, for the classification task type, we adopt LLaMA-Adapter (Zhang et al., 2023b), Large World Model (Liu et al., 2024a), VideoChat (Li et al., 2023a), VideoChat2 (Li et al., 2023b), Video-LLaMa (Zhang et al., 2023a), Video-ChatGPT (Maaz et al., 2023), Video-LLaVA (Lin et al., 2023), MiniGPT4-Video (Ataallah et al., 2024), Gemini 1.5 Pro (Reid et al., 2024) and GPT-4o [2] as baselines. Since the current models are typically employed for tasks such as text-to-video generation, with a limited number suitable for video-to-video generation, we select only NExT-GPT (Wu et al., 2023) and Open-Sora[3] as baselines for the video generation task.

---

[2] https://openai.com/index/hello-gpt-4o/
[3] https://github.com/hpcaitech/Open-Sora

Table 2: Statistics of the Physics-RW benchmark. **T1**, **T2**, **T3**, and **T4** represent tasks in mechanics, thermodynamics, electromagnetism, and optics, respectively. Class.&Gen. represent the classification and video generation task types, respectively. *Num.* indicates the number of samples included in each task type. *Sentence* indicates the average number of sentences in the instructions. *Length* represents the average length of instructions. *Frames* denotes the average number of frames in each video.

| | | T1 | | T2 | | T3 | | T4 | |
|---|---|---|---|---|---|---|---|---|---|
| | | Class. | Gen. | Class. | Gen. | Class. | Gen. | Class. | Gen. |
| **English** | *Num.* | 716 | 164 | 138 | 54 | 152 | 51 | 129 | 54 |
| | *Sentence* | 1.46 | 1.04 | 2.56 | 1.59 | 2.22 | 1.39 | 2.13 | 1.14 |
| | *Length* | 19.28 | 9.14 | 21.43 | 12.03 | 22.99 | 9.33 | 23.09 | 8.62 |
| | *Frames* | 216.0 | 427.0 | 760.7 | 662.5 | 555.8 | 522.8 | 392.9 | 312.7 |
| **Chinese** | *Num.* | 716 | 164 | 138 | 54 | 152 | 51 | 129 | 54 |
| | *Sentence* | 1.00 | 1.03 | 1.02 | 1.12 | 1.20 | 1.03 | 1.08 | 1.01 |
| | *Length* | 26.51 | 12.07 | 29.21 | 15.83 | 31.38 | 12.60 | 32.42 | 14.72 |
| | *Frames* | 216.0 | 427.0 | 760.7 | 662.5 | 555.8 | 522.8 | 392.9 | 312.7 |

## 4.2 EXPERIMENTAL SETTINGS

In our experiments, we primarily evaluate the zero-shot reasoning abilities of both open-source and closed-source models. Except for GPT-4o, we conduct experiments using the default hyper-parameters in the official code repositories, such as the number of frames for video segmentation and temperature. We observed that the frame sampling interval specified in the official GPT-4o code examples (1 frame per second) might not be universally applicable to all video lengths. Thus, we vary the frame interval based on the video length, using values of 0.5, 1, 2, and 4 seconds. This ensures a more appropriate sampling rate for videos of different durations. Given the Physics-RW benchmark's primary focus on English, we restrict our experiments to the English version. The inference of models is executed on the NVIDIA Tesla A100 and A40 GPU.

## 4.3 MAIN RESULTS

We present the performance of general world models on the Physics-RW benchmark in Tables 3 and 4. We analyze the performance of baseline models on the classification and video generation tasks in the following paragraph.

**Classification Task Type.** For the classification task, we design the instructions to prompt the model to generate "yes" or "no" as the first word of its prediction. Then, based on this first word and the corresponding ground-truth label, we calculate ACC and F1. When the output of the model does not begin with the target word, we consider it a misprediction. The experimental results are reported in Table 3. In general, three models (i.e., VideoChat2, Video-LLaMa, and Large World Model) present dissatisfied performance on four physical phenomena tasks. This is mainly because these models generate their response through a sentence rather than answering with "yes" or "no", which significantly hampers exploring their physical reasoning capabilities. To alleviate this issue, we manually review the content generated by these models, classifying them into "yes", "no", and "do not know", and then re-evaluate the performance of models. We utilize "†" to indicate that the result is obtained through human evaluation.

The experimental results show that among open-source models, Video-LLaVA and Video-ChatGPT perform the best, while GPT-4o demonstrates superior performance among closed-source models. Moreover, the performance of two closed-source models (i.e., Gemini 1.5 Pro and GPT-4o) surpasses that of open-source models across all tasks. Video-LLaMA is designed to take both video and audio as input, however, our Physics-RW does not include audio, which might further limit the performance of Video-LLaMA.

**Video Generation Task Type.** For the video generation task type, we employ instructions, (e.g., "Please predict the subsequent video"), to guide the models in generating the video. We then calculate the FVD metric based on the ground-truth videos and the generated videos. As shown in

Table 3: Comparison of models on classification tasks of the Physics-RW benchmark (%). The higher, the better. For the random guess, ACC and F1 are around 0.5. Bold text indicates the best-performing model, and underlined text shows the second-best performer.

| Models | T1 | | T2 | | T3 | | T4 | | SUM | |
|---|---|---|---|---|---|---|---|---|---|---|
| | ACC | F1 | ACC | F1 | ACC | F1 | ACC | F1 | ACC | F1 |
| VideoChat2 | 0.1 | 0.1 | 0.0 | 0.0 | 0.0 | 0.0 | 0.0 | 0.0 | 0.1 | 0.1 |
| Video-LLaMA | 0.2 | 0.2 | 0.7 | 0.7 | 0.0 | 0.0 | 0.7 | 0.7 | 1.6 | 1.6 |
| Large World Model | 0.5 | 0.5 | 2.8 | 2.8 | 0.0 | 0.0 | 4.6 | 4.6 | 7.9 | 7.9 |
| VideoChat | 15.9 | 15.7 | 20.2 | 18.9 | 18.4 | 18.0 | 25.5 | 25.2 | 80.0 | 77.8 |
| LLaMA-Adapter | 22.3 | 20.8 | 33.3 | 30.0 | 31.5 | 29.8 | 27.1 | 26.3 | 114.2 | 106.9 |
| MiniGPT4-Video | 38.9 | 31.4 | 37.6 | 31.9 | 38.1 | 31.3 | 28.6 | 25.8 | 143.2 | 120.4 |
| Video-LLaVA | 53.7 | 37.0 | 57.9 | 48.9 | 53.9 | 41.5 | 55.8 | 46.4 | 221.3 | 173.8 |
| Video-ChatGPT | 54.1 | 46.8 | 53.6 | 49.8 | 50.0 | 44.9 | 56.5 | 55.1 | 214.2 | 196.6 |
| Video-LLaMA$^\dagger$ | 2.2 | 2.2 | 3.6 | 3.6 | 4.6 | 4.6 | 5.4 | 5.4 | 15.8 | 15.8 |
| Large World Model$^\dagger$ | 39.5 | 30.0 | 34.0 | 32.1 | 41.4 | 33.3 | 32.5 | 29.2 | 147.4 | 124.6 |
| VideoChat$^\dagger$ | 27.3 | 26.6 | 38.4 | 35.9 | 33.5 | 32.7 | 37.2 | 36.9 | 136.4 | 132.1 |
| VideoChat2$^\dagger$ | 50.8 | 41.8 | 55.7 | 55.2 | 44.0 | 38.7 | 47.2 | 43.5 | 197.7 | 179.2 |
| Gemini 1.5 Pro | 55.0 | 55.0 | 68.1 | 67.7 | 60.5 | 60.2 | **58.9** | **58.0** | 242.5 | 240.9 |
| GPT-4o | **58.7** | **56.9** | **76.8** | **76.4** | **63.8** | **62.8** | 55.8 | 55.6 | **255.1** | **251.7** |

Table 4: Comparison of models on the video generation type of the Physics-RW benchmark. The lower, the better.

| Models | T1 | T2 | T3 | T4 |
|---|---|---|---|---|
| NExT-GPT | 6257 | 7207 | 4787 | 6291 |
| Open-Sora | 4516 | 5044 | 3696 | 3646 |

Table 5: Human performance on Physics-RW benchmark (%).

| Metrics | T1 | T2 | T3 | T4 |
|---|---|---|---|---|
| ACC | 82.3 | 92.7 | 90.9 | 90.6 |
| F1 | 81.9 | 92.7 | 90.9 | 90.6 |

Table 4, both Next-GPT and Open-Sora perform poorly in generating subsequent videos, indicating that there is significant room for exploration in this area.

## 4.4 HUMAN PERFORMANCE ON PHYSICS-RW BENCHMARK

To evaluate the maximum attainable performance of the Physics-RW benchmark, we selected participants who have a background in basic physics education and understand fundamental physics knowledge and conducted human evaluation. Specifically, we focused on the classification task. First, we divided the dataset for the four kinds of physical phenomena into multiple subsets. Then, we randomly assigned three humans to predict answers based on given videos and instructions for each subset. Finally, we aggregated the results from all the humans and calculated the ACC and F1 scores.

Table 5 revealed that humans consistently demonstrated strong physical reasoning abilities across all four task categories. According to the experimental results, we found that compared to the other three types of physical phenomena, human performance on mechanics tasks was relatively limited. This is mainly because the complexity of the tasks within T1 varies significantly. For example, for the dominoes subset, simpler scenarios (fewer dominoes and simpler arrangements) can be accurately predicted by most participants, but the difficulty increases substantially as the number of dominoes and the complexity of the arrangements increase. This complexity can lead to a decline in overall human performance.

## 5 ANALYSIS

To gain insights into the development of future general world models, we conducted a series of analysis experiments. In Section 5.1, we delve into experimental results and summarize the com-

mon challenges faced by current models. Sections 5.2 and 5.3 explore two potential avenues for improvement.

## 5.1 COMMON CHALLENGES IN CURRENT MODELS

**Response Biases towards "Yes".** We conducted a statistical analysis of the predicted responses from open-source models (Video-ChatGPT and Video-LLaMA) that exhibited relatively strong performance on the Physics-RW benchmark, as well as closed-source models (Gemini 1.5 Pro and GPT-4o). The analysis focused on the tendency of models to favor "yes" or "no" responses compared to the ground truth, where the number of "yes" and "no" answers was balanced.

Figure 2 illustrates the number of predicted "yes" and "no" responses using histograms. Notably, a general preference for answering "yes" is observed across all models. This tendency is particularly pronounced in the open-source models, Video-ChatGPT and Video-LLaMA, whose histograms show a clear scarcity of "no" predictions. Similarly, the closed-source model GPT-4o exhibits a significantly higher number of "yes" responses compared to "no" across all four tasks. While Gemini 1.5 Pro also demonstrates a bias towards "yes" answers in the thermodynamics and optics tasks, the difference between "yes" and "no" predictions is considerably less pronounced compared to the open-source models.

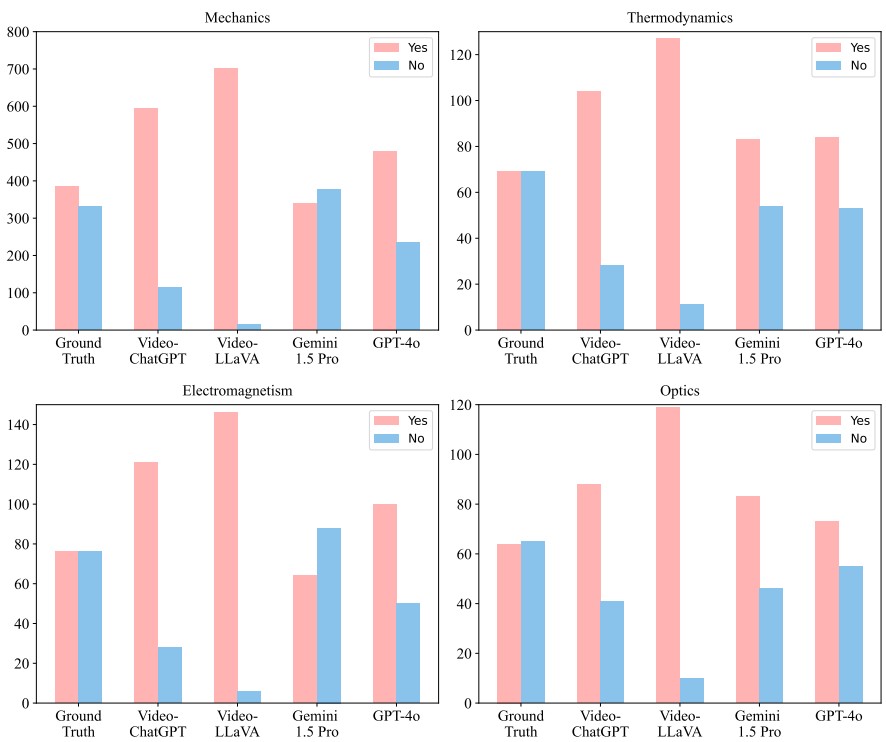

Figure 2: Tendency illustration of models to favor "yes" or "no" responses compared to the ground truth across four classification tasks in Physics-RW dataset. We count the number of "yes" and "no" responses from open-source models (Video-ChatGPT and Video-LLaMA) and closed-source models (Gemini 1.5 Pro and GPT-4o).

**Challenges with Response Format.** Although we have included explicit prompts in the instructions, such as "Answer only yes or no." or "Please answer yes or no only.", some models still generate responses that deviate from the requested "yes" or "no" format, as shown in the first video of Figure 3. Since we evaluate the models based on the first word of the generated content, these responses are classified as incorrect examples, thereby undermining the performance of the model.

**Limitations in Physical Reasoning.** As shown in the second video of Figure 3, despite the last frame of the video clearly indicating a tendency for three black dominoes to have fallen, the model

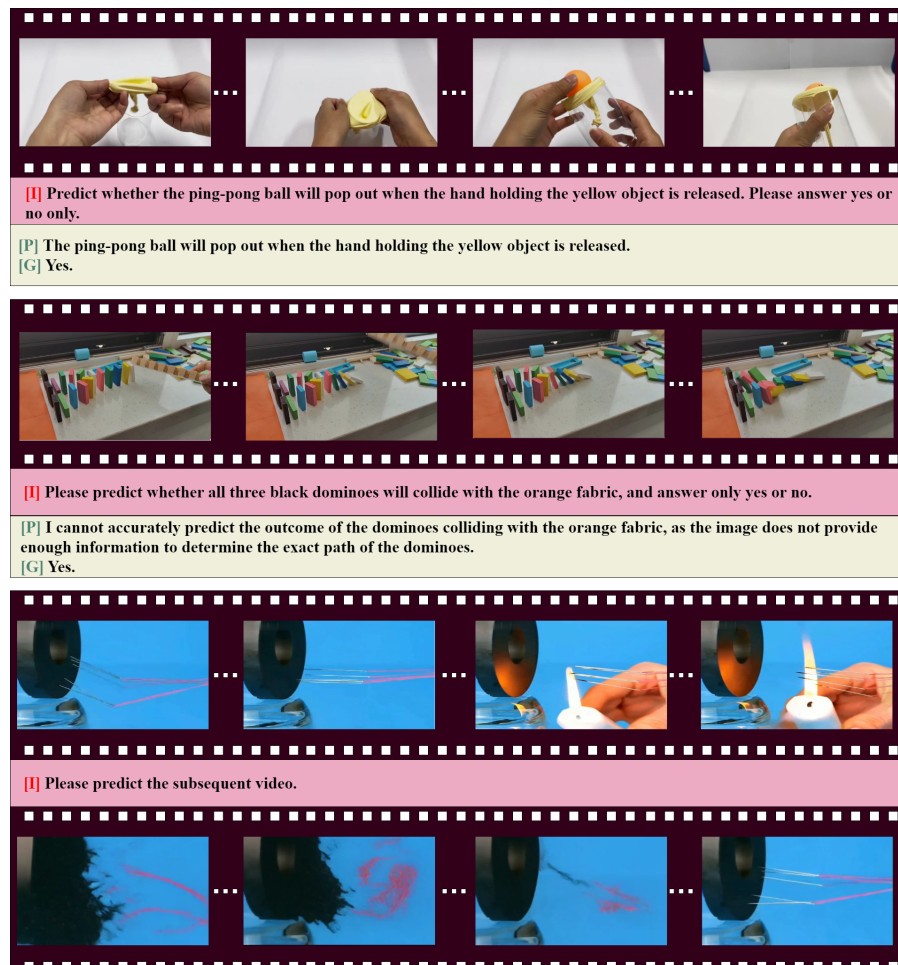

Figure 3: Examples of common challenges in existing models. [I]/[P]/[G] respectively indicate the instruction, prediction, and ground-truth.

still responds that it cannot make a prediction. This suggests that the model may struggle with interpreting the physical phenomenon, potentially due to insufficient understanding of the underlying physical principles or reasoning processes.

**Deficient Physical Understanding in Video Generation.** The last two videos in Figure 3 illustrate the models' inadequate grasp of physical phenomena in the video generation task. As shown, the model receives a video segment and an instruction as input, and is expected to generate the subsequent video based on the provided information. However, the generated video (shown last) not only deviates from a logical continuation of the input video but also contradicts physical laws. In this example, the generated video depicts a change in the shapes of the magnet and needle, which violates the principles of physics. This failure shows the current limitations of models in accurately reasoning about physical phenomena during video generation.

In summary, the understanding capability of general world models regarding physical phenomena is limited. This also indicates that there is still considerable exploration space in understanding physical phenomena.

## 5.2 POTENTIAL AVENUE I: VIRTUAL ENVIRONMENT FINETUNING

The most straightforward approach to improving the understanding ability of models on physical phenomena is to finetune the models on a large number of real-world videos. However, collecting a large number of real-world videos is extremely difficult. Fortunately, there are several simulators

and physics engines currently available that can generate a large number of virtual videos containing physics-related information. If these models can transfer the knowledge learned from virtual data to real-world scenarios, it would effectively enhance the performance of the general world models.

To investigate whether fine-tuning models within a simulated physics environment can enhance their physical reasoning abilities, we conducted experiments based on MiniGPT4-Video model. Specifically, we finetuned the models using videos related to dominoes provided in Physion (Bear et al., 2021), and then evaluated the finetuned models on a real-world domino dataset we collected [4]. We select three subsets based on the number of dominoes in contact with the target object, labeled as D1, D2, and D3. Each subset contains 100 samples.

Table 6 presents the experimental results in terms of ACC and F1 on the three test datasets. As evident from the table, models with virtual environment finetuning achieved superior performance compared to their non-finetuned counterparts (original MiniGPT4-Video). This suggests that fine-tuning on simulated data allows models to learn the underlying physical principles governing domino collisions, leading to improved abilities in inferring physical phenomena in real-world scenarios.

### 5.3 POTENTIAL AVENUE II: PHYSICAL KNOWLEDGE INJECTION VIA PROMPTS

Prompt engineering has proven effective in enhancing the reasoning capabilities of models across various domains (Wei et al., 2022; Suzgun & Kalai, 2024). Therefore, to investigate the potential benefits of injecting physical knowledge into prompts, we conducted additional experiments comparing this approach with virtual environment finetuning. Specifically, we modified the original instructions by prepending the prompt: "The collisions between dominoes are related to the laws of torque and conservation of momentum".

Table 6 presents the experimental results. Our findings suggest that while incorporating physical principles solely through textual prompts can lead to performance improvements in some scenarios (ACC in D1, ACC&F1 in D2, ACC&F1 in D3), these gains were not substantial and some scenarios (F1 in D1) have not benefited from the additional physical principles. Overall, compared to fine-tuning models within a simulated physics environment, prompting exhibited a less significant impact on model performance.

Table 6: Comparison of virtual environment finetuning and physical knowledge injection via prompts (%) on the classification task.

| Models | D1 | | D2 | | D3 | |
|---|---|---|---|---|---|---|
| | ACC | F1 | ACC | F1 | ACC | F1 |
| MiniGPT4-Video | 40.0 | 34.0 | 34.0 | 26.4 | 41.0 | 35.5 |
| w/ virtual environment finetuning | **51.0** | **49.5** | **56.0** | **54.1** | **55.0** | **52.4** |
| w/ knowledge injection via prompts | 43.0 | 30.0 | 44.0 | 33.3 | 50.0 | 36.5 |

## 6 CONCLUSION

This work presented the Physics-RW benchmark, a comprehensive physical reasoning evaluation suite designed for general world models. Encompassing real-world video and language data across four fundamental physics categories (mechanics, thermodynamics, electromagnetism, and optics), Physics-RW offers a platform for assessing models' ability to perform classification and video generation tasks based on physical phenomena. Extensive experiments revealed that current general world models exhibit relatively limited physical reasoning abilities, with significant room for improvement compared to human performance. We further explored potential avenues for improvement, including virtual environment finetuning and physical knowledge injection via prompts. We hope that Physics-RW benchmark, along with our analysis, can provide valuable insights for future research.

---

[4]We finetune the MiniGPT4-Video model based on the official script files provided: `https://github.com/Vision-CAIR/MiniGPT4-video`.

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

## A CASE STUDY

Our Physics-RW benchmark includes four tasks: Mechanics, Thermodynamics, Electromagnetism, and Optics. Each task has two types: classification and video generation. To demonstrate more erroneous cases, we provided samples of both the classification and video generation tasks for each of the four kinds of physical phenomena. Specifically, the examples for the classification task are depicted in Figure 4, while the examples for the video generation task are presented in Figure 5.

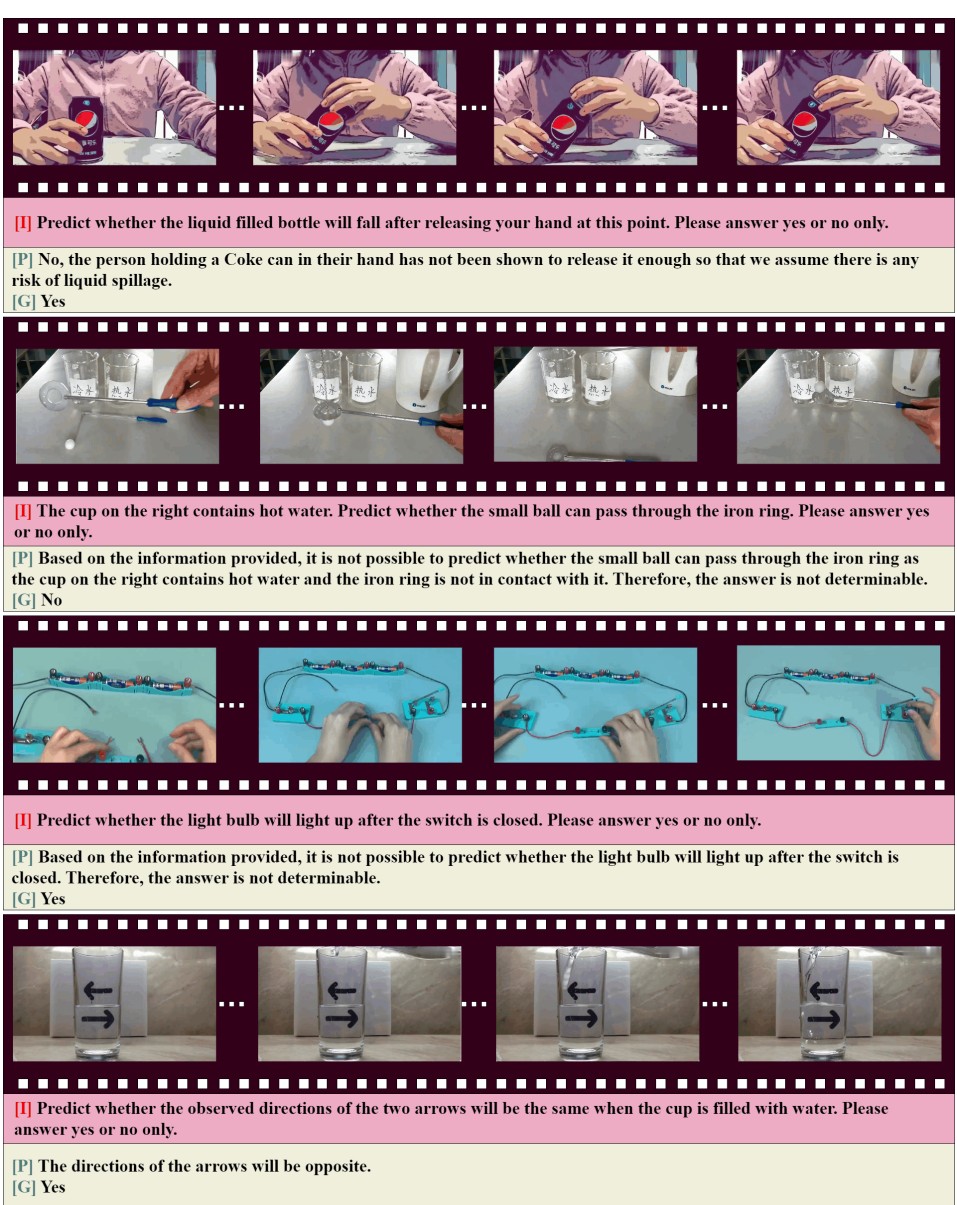

Figure 4: The erroneous cases of classification tasks in the Physics-RW benchmark, from top to bottom, are tasks related to mechanics, thermodynamics, electromagnetism, and optics. Here, [I]/[P]/[G] respectively denote the instruction, prediction, and ground truth.

## B LIMITATION AND FUTURE WORK

Our work currently faces five primary limitations.

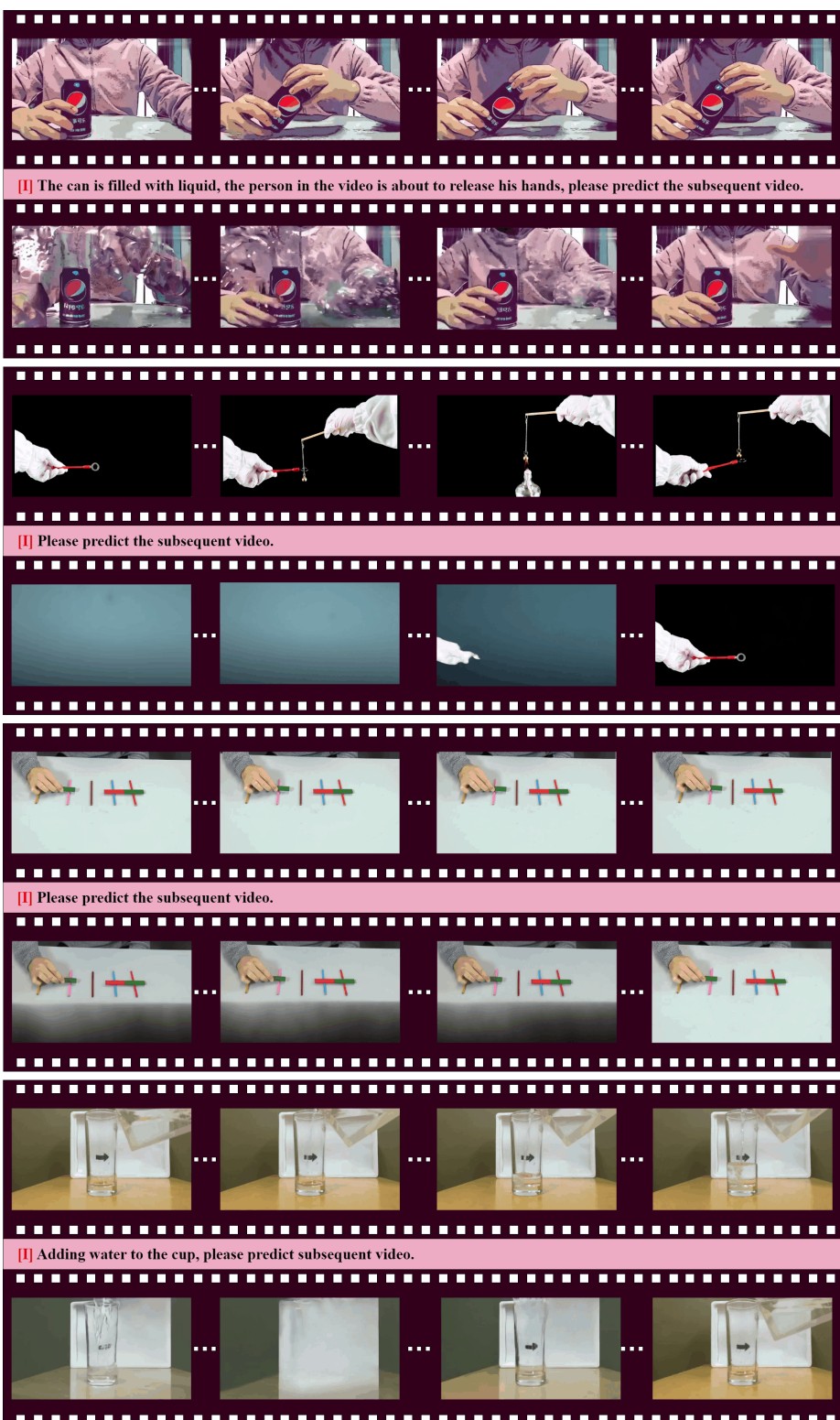

Figure 5: The erroneous cases of video generation tasks in the Physics-RW benchmark, from top to bottom, are physical phenomena related to mechanics, thermodynamics, electromagnetism, and optics. Each phenomenon includes two videos: the first video is the input to the model, and the second video is the output generated by the model. [I] represents the Instruction provided to the model.

**Data Leakage.** A significant portion of our video data originates from online sources. While this offers a diverse range of physical phenomena, it raises concerns about potential data leakage. Existing general world models might have been trained on some of the videos used for evaluation in our work. To mitigate this, we collect a new domino dataset. In future work, we will collect a wider range of video data encompassing various real-world scenarios and incorporate techniques that prevent data leakage during evaluation processes.

**Limited Exploration in Video Generation Tasks.** Our investigation into video generation tasks focuses on video-to-video models. However, the number of readily available models for this task is limited. Consequently, we have only evaluated two models. To address this, we aim to explore a broader range of video generation models in future work. This will provide a more comprehensive understanding of the effectiveness of our benchmark across different models.

**Limited Task Types in Benchmark.** Physics-RW benchmark includes two widely-used task types, i.e., classification and video generation. However, other task types (e.g., explanation) are also important for assessing physical reasoning capabilities. To address this issue, we plan to construct more forms of physical reasoning tasks in future work to achieve a more comprehensive evaluation.

**Limited Physical Phenomena in Task.** Our benchmark is primarily constructed based on the classification of classical physics, which includes five branches: classical mechanics, optics, thermodynamics, electromagnetism, and acoustics. In addition to classical physics, the realm of real-world physics encompasses many fields, such as fluid dynamics and quantum physics. However, we have chosen to focus solely on mechanics, thermodynamics, electromagnetism, and optics for two main reasons: (1) Current Model Limitations: General world models are still in their nascent stages, and their ability to handle complex physical phenomena such as acoustics (involving sound wave propagation, echoes, and reflections) is currently limited. (2) Classical Physics: The selected branches form the cornerstone of classical physics, providing a robust foundation for evaluating core physical reasoning capabilities. We believe that the current scope offers a solid foundation for evaluating the physical reasoning capabilities of general world models at their current developmental stage.

**Limited Modalities.** In the benchmark, we only utilized the language-visual modalities, without involving other modalities that could assist in physical reasoning, such as audio and tactile. This is mainly because the audio information sometimes reveals the answer prematurely, potentially biasing the evaluation of models' physical reasoning abilities. Besides, considering the current state-of-the-art general world models, most models are primarily designed to process visual information. Furthermore, incorporating audio or tactile data would introduce significant challenges in terms of data collection, annotation, and model adaptation. In future work, we plan to collect tactile videos generated when objects are touched through sensors, and combine it with the sounds produced during the interaction, thereby introducing more modalities. Additionally, we will employ the latest models in the tactile domain for evaluation.

## C  PERFORMANCE OF FINETUNING

To demonstrate the effect of fine-tuning on a specific subset, we selected training and test sets from the T1 dominoes subset for fine-tuning the Video-LLaVA model. The results are presented in Table 7. Experimental results indicate that fine-tuning on physics-related datasets effectively enhances the physical reasoning capabilities of the Video-LLaVA model.

Table 7: Comparison with finetuning-based model.

| Model | ACC | F1 |
|---|---|---|
| w/o finetuning | 50.0 | 33.3 |
| w/ finetuning | **55.5** | **45.8** |

## D  ANALYSIS OF VIDEO GENERATION METRICS

In the video generation task, we assess model performance using the common metric FVD. However, some scene changes in our dataset are relatively subtle, such as changes in the direction of arrows

(as seen in the last video of Figure 5). Therefore, we further analyze whether FVD can capture such changes. First, we extract a frame from the video and then use PS technology to reverse the direction of the arrow. Next, we copy the extracted frame and the reversed frame multiple times to construct a video. Finally, by calculating the FVD between the two videos, we can observe that due to the non-zero value of FVD, the FVD metric can reflect the changes in the arrows.

To better measure the physical visual cues of subtle changes in videos, we further propose a window-based FVD calculation method. Specifically, subtle changes typically occur in localized areas within the video. Therefore, we use a manually annotated grounding box to extract this window from the video and then calculate the FVD. This method produces FVD values that more significantly reflect minor changes compared to calculating the FVD for the entire video.

Due to the current poor performance of video generation techniques on our dataset, the FVD metric can effectively validate the model's performance. In the future, we will construct a dataset equipped with manually annotated grounding boxes to better evaluate the model's physical reasoning ability.

## E    ANALYSIS OF RESPONSE BIAS

In Section 5.1, we found that the model tends to respond with "yes". To test whether this is influenced by the prompt, we conducted an exploration based on two open-source models. We hypothesized that if the model still tends to respond with "yes" after the input prompt is reversed, this could further demonstrate the lack of physical reasoning in the general world models. We focused on the domino collision dataset from Section 5.2, where reversing the prompt can be achieved by simply adding the word "not". For instance, the original prompt "Please predict whether the red domino will collide with the orange fabric, and answer only yes or no" was modified to "Please predict whether the red domino will not collide with the orange fabric, and answer only yes or no." The number of ground-truth "yes" or "no" answers is 50 for both. We report the number of "yes" and "no" predictions made by the model in the Table 8.

Table 8: The number of "yes" and "no" predictions made by models.

| Models | D1 | | D2 | | D3 | |
|---|---|---|---|---|---|---|
| | Yes | No | Yes | No | Yes | No |
| Video-LLaVA | 100 | 0 | 100 | 0 | 100 | 0 |
| Video-ChatGPT | 100 | 0 | 99 | 1 | 99 | 1 |

According to the results, we can find that the model's predictions consistently show a strong bias towards "yes" responses, regardless of prompt phrasing. This finding further supports our claim that the models exhibit limited physical reasoning abilities.

## F    ANALYSIS OF DATASET SIZE

Our dataset contains two language versions (i.e., Chinese and English), with each version encompassing four physical phenomena. The data statistics are shown in the Table 2. The dataset size of our benchmark is comparable to similar benchmarks in the literature. For example, MME (Fu et al., 2023), a benchmark for multimodal large language models, includes several tasks with data sizes ranging from 50 to 200. JEEBENCH (Arora et al., 2023), a benchmark for evaluating reasoning abilities, covers the categories of Math, **Physics**, and Chemistry, with data sizes of 236, **123**, and 156, respectively. These comparisons demonstrate that our dataset is representative of the current state-of-the-art in terms of dataset scale for physical reasoning benchmarks to assess general world models.

Moreover, there is an imbalance in data distribution, with T1 having around seven times more data than the other tasks. This is mainly because we believe that the real-world distribution of physical phenomena should be reflected in the dataset to ensure the benchmark's ecological validity. Therefore, this imbalance primarily stems from the broader scope of classical mechanics phenomena in the real world. However, we are open to exploring potential solutions to mitigate the impact of data imbalance, for example, re-weighting techniques during model training.

## G  DATA COLLECTION AND ACCESS

**Data Curation and Preprocessing Techniques:** After acquiring the video, we first identify segments that contain specific physical events and then formulate corresponding physical questions based on the video content. We further locate the timestamps of the video frames where the answers will appear and trim the videos accordingly to ensure that the viewed content contains sufficient contextual information while excluding the answer frames. The primary bias arises from slight variations in the textual descriptions of the same question by different annotators. We ask another person for further doublechecks on the disagrees of annotations to address the variations.

**Data Storage and Management Strategies:** We will host the data on the Huggingface and ModelScope platforms, which can be accessed via the links in our GitHub repository. Besides, We will continue to update the dataset based on issues or bugs reported by users on GitHub.

**Data Release and Accessibility Plans:** Our dataset will be available through links provided on GitHub Repository and is licensed under CC-BY-NC-4.0.

**Ethical Considerations and Potential Biases in The Dataset:** The video materials we use are sourced from third-party websites. Based on the relevant licenses of third-party websites, it can be reasonably assumed that the individuals depicted in the videos are aware that their videos are intended for public access. Additionally, if anyone has objections to certain videos in our dataset, they can contact us via email, and we will promptly remove the relevant videos.

