# OpenReview forum: "Bridging the Reality Gap: A Benchmark for Physical Reasoning in General World Models with Various Physical Phenomena beyond Mechanics"
_ICLR.cc/2025/Conference — ICLR 2025 Conference Withdrawn Submission_

### Official Review · Reviewer_mZ4r · 2024-11-01

**Soundness:** 2
**Presentation:** 2
**Contribution:** 1
**Rating:** 3
**Confidence:** 5

**Summary:**

The paper introduces Physics-RW, a benchmark designed to evaluate the physical reasoning capabilities of general world models in real-world scenarios. Physics-RW includes a wide range of phenomena from classical physics, specifically mechanics, thermodynamics, electromagnetism, and optics. A zero-shot assessment of general world models on this benchmark reveals that current models exhibit limited proficiency in inferring real-world physical phenomena. The authors provide a detailed analysis of these limitations, explore various avenues for improvement, and suggest potential solutions to advance model performance in physical reasoning tasks.

**Strengths:**

The dataset’s breadth in covering varied physics phenomena is commendable, making it a valuable benchmark for evaluating diverse real-world physical reasoning.
Testing with a range of open- and closed-source models gives a comprehensive view of the state of physical reasoning in the current world model.
A comprehensive analysis of the test phenomenon is carried out, and the limitations and possible improvement methods of the model's understanding of the physical world under the current benchmark setting are summarized.

**Weaknesses:**

1. The reliance on a binary yes-or-no framework to assess models' understanding of physical rules in video content may be limiting. The prompts in Figure 1 provide substantial guidance, reducing the model’s need to reason through its decisions after making a judgment. This approach may lead to incomplete conclusions. A potentially more robust evaluation strategy could involve multiple-choice questions or require the model to justify its decisions.

2. Regarding the evaluation of video generation, the experimental scope appears limited, as a broader range of current models could be assessed. Drawing conclusions based on only two models may not provide a comprehensive view of performance across architectures. Additionally, fine-tuning open-source models on a physical reasoning dataset may be beneficial for ensuring a fair and consistent basis for evaluating open-source video generation models.

3. The mitigation strategies referenced in Contribution 3, such as additional fine-tuning and improved prompts, may not constitute a novel contribution to the paper. Fine-tuning on domain-specific datasets and refining prompts are widely accepted techniques for enhancing large language models' performance in specialized tasks and may not add unique value to the paper's contributions.

4. In the evaluation of the video generation task, the paper mentions the use of a specific prompt alongside the first half of the video to guide the model in generating the second half, with FVD employed as the evaluation metric.  This approach may be problematic, as many physical representations are inherently localized within certain regions or frames.  Consequently, FVD scores might stem from suboptimal generation in portions of the video unrelated to the critical physical interactions.  A more targeted evaluation approach could improve the focus on relevant physical phenomena, allowing for a more accurate assessment of the model's performance in generating physically consistent sequences.

**Questions:**

See Weaknesses.

---

### Official Review · Reviewer_DpN3 · 2024-11-03

**Soundness:** 3
**Presentation:** 4
**Contribution:** 3
**Rating:** 5
**Confidence:** 4

**Summary:**

This work introduces the Physics-RW benchmark, which designed to assess the physical reasoning capabilities of general world models by presenting real-world video scenarios in four categories of physical phenomena: mechanics, thermodynamics, electromagnetism, and optics. All the videos in the benchmark are real-world data. The tasks in the benchmark are split into two types—classification and video generation—evaluating models' abilities to infer or predict physical events based on video content.

**Strengths:**

- I love the idea of this paper and it's critical to evaluate the physical understanding ability of current models.
- Real-world video collected for the benchmark. It's a non-trivial effort.
- Four categories of physical phenomena are evaluated: mechanics, thermodynamics, electromagnetism, and optics.
- Both interpretive and predictive reasoning abilities are evaluated using classification and video generation tasks.
- Easy to follow. Clear writing.
- Human test is included.

**Weaknesses:**

- Some of the physical phenomena are not common in the real world life. Why does the author select these tasks?
- Missing stats of the answer of the data. Are the "yes" and "no" evenly distributed? Many models exhibited a bias toward "yes" responses in classification tasks, potentially affecting the validity of the benchmark.
- What is Zero-shot Inference in Table. 1 and why do all the previous method not support it?
- Missing video generation baselines.
- Table 3 and Table 5 can be merged.
- Some of the video is not predictable with many possibilities. How does the author select proper videos for video generation tasks?
- The dataset may contain potential data leakage, as some models may have been trained on similar data.

**Questions:**

Please see the weaknesses part.
I would like to raise my score if my concern can be solved.

---

### Official Review · Reviewer_w2hM · 2024-11-05

**Soundness:** 3
**Presentation:** 3
**Contribution:** 3
**Rating:** 5
**Confidence:** 5

**Summary:**

The paper introduces the Physics-RW benchmark, a novel dataset designed to assess the physical reasoning capabilities of general world models in real-world contexts. Unlike previous benchmarks that primarily rely on simulated videos, Physics-RW is constructed from real-world videos, spanning a comprehensive range of physical phenomena: mechanics, thermodynamics, electromagnetism, and optics. This benchmark includes two types of tasks: classification, where models infer physical properties with yes/no answers, and video generation, where models predict the continuation of physical events. Experiments using Physics-RW reveal that existing models show limited proficiency in physical reasoning, particularly in zero-shot scenarios, highlighting a need for improved physical understanding. To address this, the authors suggest possible enhancements through fine-tuning in virtual environments and the injection of physical knowledge via prompts.

**Strengths:**

- Physics-RW is the first benchmark to use real-world video data for evaluating physical reasoning across diverse physical domains (mechanics, thermodynamics, electromagnetism, optics).

- The dual-task setup (classification and video generation) allows a nuanced assessment of models’ reasoning abilities, testing both inference and dynamic understanding.

- Extensive evaluation on state-of-the-art models, with controlled factors like response format and frame sampling, ensures reliable insights into models' limitations and strengths.

**Weaknesses:**

- The paper only considers 4 categories of physical phenomena. The categories are quite limited and only cover a small amount of phenomena in the real world. I suggest the authors include more physical phenomena such as fluid and chemical reactions.

- The authors may provide a detailed analysis to explain why many models perform worse than random guesses in Table 3

- FVD may not be a good metric for evaluating the physical similarity of ground truth and generated videos. FVD is more of a semantic similarity metric. The authors need to justify their choices of using FVD as evaluation metrics.

- The authors need to provide deeper insights about why current VLMs are not good at physical reasoning. For example, is training data the main reason for this observation?

- The authors need to provide more baselines for the video generation task, e.g. stable video diffusion, etc.

**Questions:**

- What is the size of the proposed dataset? Is this significantly larger than previous datasets and benchmarks?

- Is there a clear timeline for releasing the benchmark? Is there any document and anonymous website for this benchmark?

---

### Official Review · Reviewer_YX57 · 2024-11-07

**Soundness:** 2
**Presentation:** 2
**Contribution:** 2
**Rating:** 5
**Confidence:** 4

**Summary:**

This paper proposes the Physics-RW benchmark to measure the understanding of physical laws by video understanding models and video generation models. It conducts experiments on some open-source and closed-source models and analyzes the results.

**Strengths:**

The authors established a relatively comprehensive dataset to measure the understanding of physical laws by video understanding models and video generation models, analyzed the results in detail, and designed certain experimental explorations to improve the model.

**Weaknesses:**

- line 099 "Encompassing the four major categories xxx" , why you choose this four categories?
  - line 103 “xxx generate subsequant videos", Why did the author choose to model it as a video2video task? Can't text2video measure the physics understanding ability of models like sora? The author's previous motivation started with sora, but this is a T2V model, and now a V2V benchmark is designed. I don't understand the logic here.
  - line 138: some papers like VideoPhy, VideoScore, PhyGenBench already evaluate capabilities of video generation models to simulate the world。

  - line 193: In order to check whether the generated video conforms to the laws of the physical world, it is not enough to just use FVD. This is explained in VideoPhy/PhyGenBench, so it is not reasonable for the author to only use FVD to judge the video quality as a criterion for measuring the ability of video generation models to simulate the world. In other words, if the author designs for understanding the laws of physics, this is not to only evaluate video quality.
  - line 259: “we manually review..." , In some cases, the author manually evaluates the response of the model, which makes the results difficult to reproduce and makes it difficult for others to use this benchmark. Why not use tools like VLMEvalKit?
  - The author said that the results in Table 4 show that the current model lacks understanding of physical laws, which is an overclaim. Because most of the training tasks of video generation models are T2V, and V2V itself is even more difficult, this will affect the author's evaluation of the correctness of physical laws.
  - line 329，As shown in Figure 2, many models tend to answer yes directly (Video-LLaVA), which is similar to cheating and will lead to errors in the evaluation results. The author should design certain robust evaluation methods to avoid misjudgment caused by the model always answering yes/no. For example, for a question, ask its affirmative description (the answer is yes) and negative description (the answer is no) at the same time, and both must be answered correctly to be considered correct.
  - I wonder if the author has explored the experiment of using Video-VLM to extract the caption of the video and then give it to LLM for Yes/No QA judgment. I think using the common sense of LLM should be able to achieve a significant improvement on this benchmark.

**Questions:**

Please refer to details in weaknesses.

---

### Official Review · Reviewer_rT6t · 2024-11-08

**Soundness:** 3
**Presentation:** 2
**Contribution:** 3
**Rating:** 3
**Confidence:** 4

**Summary:**

This work introduces a new benchmark Physics-RW to evaluate the real-world physical reasoning ability of the world model. The author points out existing evaluation methods rely on simulated video data, which makes it difficult to fully reflect the physical reasoning ability of the model in the real world. The Physics-RW is constructed through real-world videos and covers four major categories of physical phenomena: mechanics, thermodynamics, electromagnetism, and optics. Moreover, the paper also compares the results of existing models and methods, such as fine-tuning the virtual environment and injecting physical knowledge through prompts to improve performance. The main contribution of this work includes its benchmark, experiment, and demonstration methods.

**Strengths:**

1. The giving problem is obvious but important “Can general world models perform accurate physical reasoning across various phenomena?”.
2. This paper notices the research gap in physical understanding and collects the videos from the real world for evaluation and comparison.
3. The author separates the task into four domains (mechanics, thermodynamics, electromagnetism, and optics), the major categories of classical physics.
4. This benchmark includes video understanding and generation, extending the physical reasoning ability to the multimodal domain.

**Weaknesses:**

1. Though the target general question is clear, the problem definition is confusing. In the abstract, sometimes it is "video understanding and generation", or "reasoning ability", while in Sec 3.2, it goes like "As our benchmark incorporates two task types (i.e., classification and video generation)".  I suggest the author use the unified expression.
2. A similar question exists in the "world model", what is the world model? What is the relation between world model and video understanding, segmentation, generation, or classification...
3. “T1, T2, T3, and T4 represent tasks in mechanics, thermodynamics, electromagnetism, and optics, respectively” I would suggest the author use a better abbreviation since T1T2 is hard to follow while reading.
4. In the experiment, the author compares the " general world models ", while evaluating the VLM and video generation. A comprehensive explanation would reduce the confusion.
5. : Comparison of models on the video generation is weak compared to the classification tasks. How are other open-source video generation models performing?
6. As a benchmark for a specific domain, more comparison and analysis of human evaluation is required. Is the human evaluation result aligned with the quantitative result? Can your benchmark be good enough to showcase the question you giving in the beginning? How future research can use your benchmark while the human evaluation can not be repeatable?

**Questions:**

Please refer to the weakness part. I include the questions there.

---

### Official Review · Reviewer_8J5e · 2024-11-09

**Soundness:** 3
**Presentation:** 2
**Contribution:** 2
**Rating:** 5
**Confidence:** 3

**Summary:**

This paper proposes a new real-world physical reasoning benchmark including four common physics categories ((mechanics, thermodynamics, electromagnetism, and optics). Classification task and Video generation task are included in this benchmark, enabling the benchmark to verify both the physical reasoning ability and generative modeling capability of models. The authors conduct zero-shot experiments across several world models on the benchmark, exploring several avenues for improvement.

**Strengths:**

1. The benchmark is extensive, covering a range of physics categories, allowing for a thorough assessment of the performance of different world models.

2. The authors carry out both zero-shot and fine-tuning experiments on multiple world models, showing significant potential for models to advance in physical reasoning.

3. It is intriguing that world models tend to respond with "yes" more often.

**Weaknesses:**

1. The classification task of answering "yes" or "no" seems overly simplistic. Adding a broader range of tasks, such as multiple-choice or fill-in-the-blank questions, could significantly enhance the benchmark.

2. Some of the videos appear to require additional review and adjustment. For instance, in Figure 1, example [C] of Optics, it’s unclear from the first frame whether the word "won" is on the front wall of the glass, the back wall, or in the background, each of which would lead to completely different interpretations.

3. It's puzzling that the accuracy metric (ACC) for most world models on the classification task is below 50%, especially given that the dataset has a balanced distribution of "yes-no" answers. Could this result from a high number of " do not know" answers in the models, or might there be another cause? I hope the authors will carefully discuss this phenomenon.

**Questions:**

I'm curious about the authors' criteria for assigning videos to either the classification or video generation tasks. Since all videos could technically be used in the video generation benchmark with the addition of specific instructions, why does Table 2 show a much larger number of videos for the classification task than for the generation task?

---

### Note · Authors · 2024-11-18

I have read and agree with the venue's withdrawal policy on behalf of myself and my co-authors.